# The Impact of the COVID-19 Pandemic on Rural Food Security in High Income Countries: A Systematic Literature Review

**DOI:** 10.3390/ijerph19063235

**Published:** 2022-03-09

**Authors:** Katherine Kent, Laura Alston, Sandra Murray, Bonnie Honeychurch, Denis Visentin

**Affiliations:** 1School of Health Sciences, Western Sydney University, Campbelltown 2560, Australia; 2School of Health Sciences, University Tasmania Launceston, Tasmania 7250, Australia; sandra.murray@utas.edu.au (S.M.); bonnie.honeychurch@utas.edu.au (B.H.); denis.visentin@utas.edu.au (D.V.); 3Faculty of Health Deakin Rural Health, Deakin University, Warrnambool 3280, Australia; laura.alston@deakin.edu.au; 4Institute for Health Transformation, The Global Obesity Centre Deakin University, Geelong 3220, Australia

**Keywords:** rural, food security, high-income, COVID-19, food access, food availability

## Abstract

Prior to the COVID-19 pandemic, rural-dwelling people in high-income countries were known to have greater challenges accessing healthy food than their urban counterparts. The COVID-19 pandemic has impacted food supplies across the world, and public health restrictions have changed the way people shop for food, potentially exacerbating food insecurity. This systematic literature review aimed to synthesize the available evidence on the impact of the COVID-19 pandemic on aspects of food insecurity in rural populations residing in high-income countries. Five electronic databases were searched, identifying 22 articles that assessed food insecurity prevalence or data on food availability, access, utilization and the stability of the food supply in rural populations during the COVID-19 pandemic. Ten studies examined the prevalence of food insecurity in rural populations, with the reported prevalence ranging from 15% to 95%. Where rural/urban comparisons were presented, most studies (*n* = 5; 71%) reported that food insecurity was significantly higher in rural regions. Five studies examined the availability of food and eight studies examined access to food, identifying that rural populations often had lower food availability and access to food during the pandemic. In contrast, two studies identified positive effects such as more gardening and increased online access to food. Rural populations experienced multiple changes to food utilization, such as reduced diet quality and food safety observed in eight studies, but this was not shown to be different from urban populations. Additionally, the food supply in rural regions was perceived to be affected in two studies. The results of this review may be used to inform region-specific mitigation strategies to decrease the impact of the current COVID-19 pandemic and future global events on food security. However, the lack of consistency in study outcomes in research on rural populations limits the identification of priority areas for intervention at a global-scale.

## 1. Introduction

Food security is said to exist when all people, at all times, have physical, social and economic access to sufficient safe and nutritious food that meets both their dietary needs and food preferences for an active and healthy life [1]. Food security is multi-dimensional, encompassing four pillars: (1) food availability, which corresponds broadly to the availability of food in the food supply; (2) food access, encompassing both physical and financial access to food; (3) the utilisation of food, namely the processing and consumption of food, which is related to diet quality; and (4) stability, which refers to the stability of the three other pillars.

Food insecurity, which reflects the absence of the aforementioned pillars, has adverse health and social effects across the lifespan, as it contributes to poor diet, resulting in higher rates of chronic diseases such as obesity and cardiovascular disease (CVD). Poorer diet has been shown to contribute to the gap between health in metropolitan and rural areas, with one study showing that >4000 CVD deaths could be prevented annually in rural Australia alone if rural communities were able to achieve healthy diets [2]. While chronic diseases disproportionately affect rural populations [3], unfortunately most of the data sources describing the prevalence of food insecurity do not distinguish populations in rural and remote locations from more densely populated areas. When it is examined, some studies report the prevalence of food insecurity as being greater in rural regions in some high-income countries. For example, a study in the USA showed that food insecurity was higher in rural regions (17.7%) compared to urban regions (14.5%) [4]. Furthermore, in 2012-13, approximately 20 percent of rural-dwelling Aboriginal and Torres Strait Islander Australians were food insecure [5]. However, in other studies, the prevalence of food insecurity has been reported to be similar between rural and urban communities, which may reflect differences in sociodemographic characteristics and food access in different rural regions. For example, the prevalence of food insecurity was not significantly different in urban (12.4%) and rural (10.3%) regions in Canada [6], or in a focussed study in Wisconsin, USA (urban core, 14.1%, other urban areas, 6.5%, and rural areas, 10.5%). Ultimately, the burden of food insecurity and nutrition-related outcomes in rural populations is under-researched in other high-income countries [7], and further research to uncover the prevalence of region-specific food insecurity remains a priority.

Research related to food access, food availability and the utilisation of food more consistently shows that rural regions in high-income countries have poorer outcomes compared with urban regions. In particular, there are additional barriers to obtaining food in rural areas, such as the distance required to travel to obtain food and transportation issues [8]. In addition, there is a lower availability of food [9,10,11], and poorer promotion and higher prices of foods [12]. Previous studies in high-income countries have also highlighted the lower quality of food aid and poorer access to services that address food insecurity in rural regions [13]. Consequently, there is evidence to suggest that food insecurity, including the limited access to, availability, and utilisation of food, contributes to rural health inequalities even in high-income countries [14].

The COVID-19 pandemic is continuing, with new variants arising around the world on a regular basis, providing ongoing challenges for high-income countries. Since early 2020, the COVID-19 pandemic has been associated with numerous public health restrictions variably applied in countries across the globe, including physical and social distancing, travel restrictions, and closure of non-essential services. These restrictions have been important to slow the rate of transmission of COVID-19 and protect health services, and have shown to be beneficial in reducing mortality, improving welfare and be economically viable in high-income countries [15]. However, these restrictions have negatively impacted the availability, access to, and utilisation of food, and have also affected food supply stability [16]. The impact of the COVID-19 pandemic on food insecurity and related outcomes in low and middle-income countries has been reviewed and published [17,18]. However, the impact on food insecurity in high-income countries remains unclear. In many high-income countries, it was suggested that rural communities were more protected than urban regions in the first wave of the COVID-19 pandemic due to their remoteness and lower population density. However, when outbreaks of COVID did occur in rural communities in high-income countries, they were more impactful [19,20], as they typically have older populations with a higher burden of chronic disease, in addition to poorer access to intensive care, resulting in higher mortality risk. Furthermore, persistent and severe food supply shortages resulting from panic buying and transport/logistics issues throughout the food supply have had the potential to impact household and community-level food security disproportionately in rural communities with fewer food purchasing options [21].

To inform appropriate initiatives and policy changes to support food security in rural areas of high-income countries throughout the COVID-19 pandemic and beyond, an understanding of how the COVID-19 pandemic and associated physical and social distancing restrictions impacted food security in these communities is needed. Therefore, we sought to synthesize the evidence from published studies that examined the impact of the COVID-19 pandemic on food security in rural communities in high-income countries. The aim of this systematic review was to synthesize the evidence of how the COVID-19 pandemic impacted the prevalence of food insecurity and the pillars of food access, availability, utilisation and stability in people residing in rural and remote areas of high-income countries to identify strategies for potential intervention and to make recommendations for future research.

## 2. Materials and Methods

### 2.1. Protocol and Registration

A protocol for the review was submitted to the International Prospective Register of Systematic Reviews (PROSPERO; ID 296790). This systematic review was conducted and reported following the Preferred Reporting Items for Systematic Reviews and Meta-Analysis (PRISMA) statement guidelines [17]. The Appendix A includes a PRISMA checklist for this article (Appendix A).

### 2.2. Eligibility Criteria 

This review aimed to synthesize information about the impact of COVID-19 on food security, including the prevalence and the domains of availability, access, utilisation, and stability in rural regions of high-income countries. For ease, the term ‘rural’ is used throughout this paper; however, definitions of rurality differed between study settings. Studies were included according to the inclusion and exclusion criteria outlined in Table 1.

### 2.3. Information Sources and Searches

Five major electronic databases (CINAHL[EbscoHost], Academic search complete [EbscoHost], Scopus, Medline [Ovid], Web of science), were systematically searched on 4 December 2021. Search terms included combinations, truncations and synonyms of the following:Food Security: “Food Securit *” OR “Food Insecurit *” OR hunger OR “apparent consumption *” OR “food *”.COVID: Coronavirus OR COVID* OR Pandemic * OR SARS-CoV-2.Other food security outcomes: access * OR hoarding OR availabil * OR pric * OR income * OR prepar * OR skill * OR suppl * OR environment * OR “panic buy *” OR diet * OR fruit * OR vegetable * OR dairy * OR meat *.

### 2.4. Study Selection

One author carried out all electronic database searches. Search results from individual databases were added to the Covidence systematic review software (Veritas Health Innovation, Melbourne, Australia). Duplicate records were automatically removed. Study selection followed the process described in the Cochrane Handbook of Systematic Reviews [22]. While all authors contributed to screening, a minimum of two authors independently screened all titles, abstracts and full text articles against the eligibility criteria to remove irrelevant studies. Conflicts at each stage of the screening process were discussed and resolved by consensus between two authors, and when required, further consultation with the whole project team was sought.

### 2.5. Data Collection Process

Studies that met the inclusion criteria after the full-text screening stage underwent data extraction. Data were extracted using a pre-designed electronic data extraction table that included details such as author, year of data collection, population, number of participants, data collection methods, food security outcome measure, findings related to the rural area and key findings of the study. One author extracted the data, and two authors conducted an independent cross-check of all studies for accuracy. When data were unclear, the authors contacted the corresponding author of the manuscript.

### 2.6. Data Synthesis and Analysis

Data from the included studies were summarised in a narrative synthesis and tabulated using the information collected from the data extraction form. The primary outcome of interest was food security prevalence, or any outcome that examined impact to the food security pillars of food availability, access, utilisation or stability which could be used to characterise food insecurity in rural areas.

### 2.7. Quality Assessment

The quality of included studies were appraised using Joanna Briggs Critical appraisal tools. Two reviewers assessed the quality of the studies using the JBI Critical Appraisal Checklist for Analytical Cross Sectional Studies, the Checklist for Qualitative Research [23], and studies with the primary aim of determining food insecurity prevalence were appraised using the Checklist for Prevalence Studies [24].

## 3. Results

### 3.1. Study Selection

As summarised by the PRISMA diagram in Figure 1, the searches across the five databases retrieved 12,593 abstracts. After the removal of duplicates, a total of 7080 articles underwent title and abstract screening. Of these, the full texts of 329 articles were reviewed, resulting in 22 articles meeting the inclusion criteria. The main reasons for exclusion at the full-text stage included that the studies: did not include a rural population or stratify their outcomes according to rurality (*n* = 197), were conference abstracts (*n* = 41), or had incorrect study design (e.g., case studies, opinion pieces or reviews) (*n* = 44), did not examine the correct outcomes (*n* = 14), or were not related to COVID-19 (*n* = 7). Three full text documents were excluded as the abstract was in English, but the full text was not. Lastly, one study was excluded as although it reported food insecurity prevalence in a rural sample [25], the prevalence had been reported in an another included study by the same author, and the study did not stratify other food security results (such as food access and food availability) according to rurality.

### 3.2. Description of Studies

Full details of the study designs, settings, populations, and domains of food security examined are summarized in Table 2. More than half of the 22 included studies were conducted in rural USA (*n*= 13) [26,27,28,29,30,31,32,33,34,35,36,37,38], with others being conducted in Australia [39,40], Canada [41], the UK [42], Poland [43], Israel [44], Chile [45] and Germany [46]. One study was conducted in an international sample, comprising participants from 62 different countries [47]. Most studies (*n* = 16) collected data on adult populations or household-level information [27,29,30,31,34,35,36,37,38,39,40,41,42,43,44,45,47], while two specifically focused on households or families comprising both adults and children [26,32]. Three studies reported on the impact of COVID-19 on food outlets such as stores [28], farmers markets [33], and food banks [46]. Sixteen were quantitative in design [26,28,29,30,31,32,33,37,38,39,41,43,44,45,46,47], with a sample size for the rural sample ranging from 50 to 1766 (Table 2). Three studies were mixed-methods [27,34,42] and three were qualitative [35,36,40], with the rural sample size ranging from 14 to 92 (Table 2). Of the included studies, most (*n* = 12) captured food insecurity-related data using online surveys [26,27,29,30,34,37,38,39,41,42,43,46,47]. Others utilized telephone surveys [33,44,45] or surveys conducted via text-messages [32]. Three studies collected data using focus groups [35,40] or interviews [36]. One study collected data directly from participants through their standard care appointments with a community service program [31]. One study utilized the mapping of travel data in the vicinity of grocery stores [28].

### 3.3. The Prevalence of Food Insecurity

An overview of the food security outcomes extracted from studies reporting on food insecurity prevalence statistics in rural areas is presented in Table 3. Eight quantitative studies [26,30,32,37,38,39,41,45] and two mixed-methods studies [27,34] reported on the prevalence of food insecurity in their study sample. The prevalence statistics reported in all of the included studies were determined using a variety of validated instruments. Four studies utilized the U.S. Household Food Security Survey Module: Six-Item Short Form [26,37,38,39], two used the Hager two-item screening tool [27,34], with one study each using the UN FAO Food Insecurity Experience Scale [45], a variation on the Current Population Survey Food Security Supplement [32], and the 2-item Hunger Vital Sign screening tool [30]. Finally, one study used a six-item questionnaire adapted from an 18-item questionnaire that is routinely used to monitor 12-month food insecurity in Canada [41]. The recall periods for these validated tools ranged from ‘daily’ to 30 or 90 days. Other studies also used a non-specific time period to capture experiences only during the pandemic (e.g., “since the coronavirus outbreak”) (Table 3).

While many studies addressed rurality in their assessment of food insecurity, the prevalence of food insecurity for rural participants or a comparison to a non-rural subgroup was sometimes not reported. Where possible, the authors of this review used the study data to calculate the food insecurity prevalence for these subgroups and perform proportion tests between them. Where these data are reported in Table 3, they are indicated by bold and italics. Of the ten studies reporting on the burden of food insecurity in rural areas, three reported statistics in a sample comprising only rural-dwelling people [32,34,37], while seven compared rural and urban dwelling participants [26,27,30,38,39,41,45]. The prevalence of food insecurity was reported to be 14.9% and 15.3% in rural dwelling households in Canada [41] and the USA [26], respectively. This was lower than the prevalence of 25% and 29% found in studies of the rural state of Vermont [37,38]. A third (33%) of rural respondents in Tasmania, Australia were classified as food insecure [39], which was similar to the 40% and 40.5% of participants experiencing food insecurity in two studies of adults in the USA [27,30]. It was also reported that 38.7% of families with children were worried about running out of food after school closures in the USA [32]. The prevalence was higher in rural Chile, with more than half experiencing food insecurity (53.5%). The highest prevalence was documented in a sample comprising only rural-dwelling people who were participants of a rural emergency food program (95%) [34].

Of the seven studies reporting a comparison between rural and urban populations, five studies reported that food insecurity was significantly higher in the rural dwelling population (15.3% rural vs. 14.5% urban [45]; 33% rural vs. 23% urban [39]; 14.9% rural vs. 11.3% urban [26]; 25.1% rural vs. 21.6% urban [38]; 53.5% rural vs. 45.9% urban [41]) with the food insecurity in the rural population ranging from 3.5% to 11% higher than their urban counterparts. No significant difference between rural and urban participants was reported in one study [27], and while one study showed a difference between rural and urban populations, but this was not found to be significant after adjusting for several demographic characteristics in a multivariate model [38]. One study reported that food insecurity was higher in urban areas (62.3%) compared with both rural (40.5%) and suburban areas (36.7%) [30].

Additionally, one study reported statistics both before and during the COVID-19 pandemic [45]. This study of pre- versus during COVID-19 showed that the prevalence of food insecurity for rural Chileans increased from 29.9% pre-COVID to 53.5% during the pandemic [45]. Another study showed that 9.1% of rural participants were newly food insecure during the COVID pandemic [38].

### 3.4. Food Availability

An overview of the included studies reporting on the domains of food availability is presented in Table 4, including four quantitative studies [30,33,38,46] and one qualitative study [40]. All these studies reported that the COVID-19 pandemic impacted on the availability of food in all settings. In four studies, a rural and urban comparison was performed to determine differences in the availability of food [30,33,38,46]. Two of these studies showed that there was no difference in study outcomes according to rurality [38,46], and two studies showed that urban areas were more greatly impacted [30,33].

A study of food service managers and consumers in Australia reported that food was unavailable in rural areas due to consumer panic buying and issues with the food supply system resulting from the necessary public health restrictions at the beginning of the pandemic [40], which was a cause for worry in the study participants.

Another study in a USA sample directly surveyed people living in rural areas to determine if food unavailability was an issue for them during the COVID-19 pandemic [38]. While issues with the availability of food affected many food-insecure households during the pandemic in this study, rural dwelling respondents did not report any difference in the availability of food compared to urban dwelling respondents [38].

One study reported that the availability of food for vulnerable people was reduced through the closure of food banks at the beginning of the pandemic, but the lengths of closures were similar in rural and urban areas [46]. Conversely, in a study of farmers markets [33], rural areas were less affected than urban areas, in that they were more likely to make a profit during the pandemic, and also lost fewer staff and patrons throughout this time.

### 3.5. Food Access (Physical and Financial)

An overview of the included studies reporting on the domains of food access is presented in Table 5. Five quantitative studies [28,30,31,32,38], three qualitative studies [35,36,40] and one mixed-methods study [34] investigated challenges related to access to food in rural areas during the pandemic. All of the studies highlighted that there were various challenges in accessing food during the pandemic.

Where rural and urban differences were compared, the findings were mixed. For example, one study in Vermont, USA reported that food-insecure households faced numerous challenges in accessing and affording food in the pandemic compared with those who were food-secure, but this was not associated with rurality [38]. In another study, transportation as a barrier to food access did not vary significantly by rural or urban areas [30]. Conversely, one study of traffic in the vicinity of shops in rural and urban areas indicated that urban shops had significantly more reduced traffic compared with rural areas, indicating that access to food may have been less affected in the rural regions [28]. Similarly, another study reported that there were more barriers to accessing food for those living in an urban area than those in rural areas [30].

One study reported that access to food in a rural Australian community was reduced due to widespread panic buying of food, and that the shops were unable to restock the shelves quickly [40]. In another study in a rural community in the USA, it was reported that a major recurring issue in their interviews was the perceived scarcity of food and increase in food costs during the pandemic, which poorly impacted food access. However, food security was a persistent issue in this community, which was exacerbated by both the COVID-19 pandemic and other disasters.

In three studies, emergency food relief agencies were reported to provide an important service for accessing food some rural communities during the pandemic [31,32,34]. One study showed that families who experienced food insecurity during school closures were more likely to rely upon and benefit from emergency food relief packs. In another study, the use of emergency food relief in rural areas was reported to have increased when compared to prior to the pandemic [31]. The reverse relationship was reported in another setting, as some existing services that provided emergency food assistance were closed during lockdown [35].

One positive impact of the COVID-19 pandemic on food access was reported in a rural study with the rapid implementation of online food shopping, which had not been available in these areas before the pandemic [35].

### 3.6. Utilization (e.g., Cooking, Storage)

Five quantitative [32,38,43,44,47], one qualitative [35], and two mixed methods studies [27,42] identified changes in the utilization of food throughout the COVID-19 pandemic in rural areas (Table 6), spanning the concepts of food safety, stockpiling food, eating behaviours, and cooking behaviour.

Two studies examined safety and fears regarding contracting COVID-19 from shopping or from food sources, which were increased during the pandemic but were not associated with rurality, and hence rural and urban areas were similarly affected [38,43]. Another study examined self-reported behaviour around stockpiling food, which shows that rural adults were less likely to report stocking up on food than adults living in urban regions [44].

Changes in food choices, consumption behaviour, and diet quality were observed in five studies. One study in a rural sample reported that approximately a quarter of the sample reduced their consumption of fruits, vegetables and red/processed meat during the pandemic [37]. Furthermore, another study reported that living in a rural area was associated with a decrease in the diversity of vegetable consumption [47]. In the other three studies, there were similar impacts on diet quality and food choices which were similar for both rural and urban areas [27,42,43].

Finally, one study reported positive food utilization behaviours in a rural sample, such as increased time spent cooking, a greater proportion of food being consumed at home, and increased produce gardening [35].

### 3.7. Stability of the Food Supply

One quantitative [29] and one qualitative study [40] measured respondents’ perceptions of the stability of the food supply (Table 7). In both studies, the perception of the food supply was that it had been negatively impacted by the COVID-19 pandemic. Whelan et al. [40] described the issues with food availability and access as the food supply was unable to re-orient itself due to large increases in consumer demand for food in rural Australia. The other study on low-skilled workers in the USA reported that participants perceived that agricultural production was more important than prior to the pandemic and there was increased concern of food shortages, but this was not related to rurality.

### 3.8. Quality Assessment

Four studies had the primary aim of measuring the prevalence of food insecurity in a population, and were therefore assessed with the prevalence tool [26,38,39,45]. A further fourteen studies were appraised using the cross-sectional tool [27,28,29,30,31,32,33,37,41,42,43,44,46,47]. The remaining four studies were appraised using the qualitative tool [34,35,36,40]. The results of the critical appraisals are available in Appendix A. The majority of studies included clear information and documented methodological detail to enable quality assessment.

## 4. Discussion

The aim of this review was to synthesize the available evidence investigating the impact of the COVID-19 pandemic on food insecurity in rural populations residing in high-income countries. Across high income countries, there were 22 studies that met the inclusion criteria and examined a broad range of outcomes. Most of the included studies showed the prevalence of food insecurity and outcomes related to the four dimensions of food security have been affected by the COVID-19 pandemic and associated public-health restrictions in rural areas, but the outcomes varied across different regions.

All studies reported a high prevalence of food insecurity in rural areas, although there was large variation in the levels found in different settings. The study designs, populations, and instruments used to assess food insecurity varied greatly across the studies, making direct comparisons difficult. When a rural/urban comparison was conducted, most included studies reported that the prevalence of food insecurity in rural areas was higher than in urban-dwelling populations during the pandemic. Not surprisingly, this indicates that rural-dwelling people have been disproportionately impacted by the COVID-19 pandemic, and it is possible that existing rural/urban inequalities have been amplified by COVID-19. Some research has suggested that rural regions in high-income countries may be more susceptible to negative impacts of the pandemic, due to sociodemographic differences such as lower income, in addition to lower social capital, higher rates of chronic diseases, and poorer delivery and access to social services in these areas [48]. Additionally, people in rural areas may also be more vulnerable to food insecurity due to lower rates of charitable giving and fewer community civic organizations [48], which may limit the resilience and local response of rural communities to the pandemic. The studies also varied in the time-point and local situation with regard to the pandemic-related restrictions including lockdown measures, and these were often not clearly described in the included studies. Furthermore, the included studies often did not comprehensively examine the primary causes of food insecurity. Where this was identified across the studies, it was predominantly attributed to increases in food prices, food shortages, and issues with social distancing restrictions implemented during the pandemic. While rural regions were initially believed to be protected against high rates of infection compared to urban regions at the beginning of the pandemic, a transition to higher infection rates and mortality in rural areas was documented as the pandemic progressed [49,50]. Substantial negative impacts on unemployment and economic outlook in rural regions have also been reported compared to urban populations [51]. While all studies were either in a rural setting or assessed a rural subpopulation, not all studies reported the prevalence of food insecurity for the rural group, and some did not provide a comparison in the prevalence between the rural and non-rural participants. The variation in reporting across the studies, in part due to their different aims, limits the ability to use the review results to further synthesize the prevalence of food insecurity in rural populations across high-income countries and to assess the difference from urban populations.

Our review shows there was substantial variation between the food insecurity prevalence statistics of included studies, which may relate to differences in the spread of the outbreak or the pre-pandemic prevalence of food insecurity, in addition to differences in the public health restrictions, including government-ordered closures in hospitality businesses and food outlets, or disparities in levels of unemployment and income changes through the pandemic [51]. In other studies, food insecurity was not substantially different between rural and urban areas, which indicates that there is a substantial economic impact of the pandemic on food security for both rural and urban areas in high-income countries. Only one study showed that urban areas experienced higher levels of food insecurity than rural and suburban areas, which could relate to differences in sociodemographic characteristics in this region or increased vulnerability to supply chain issues due to lower levels of urban agriculture and home food production compared with rural regions [52]. 

High-income countries reported surges in demand for food and supply chain disruptions at the beginning of the COVID-19 pandemic [53], which reduced the availability of food. While these issues are not isolated to rural regions, the impacts of lower availability of food are potentially worse for rural people, many of whom struggled with the availability of food prior to the pandemic due to geographical barriers. For example, it was reported that rural people were significantly more likely to perceive their food environment as having an inadequate number of supermarkets prior to the pandemic [54]. Furthermore, food supply systems may be slower to rebound in rural regions as infection rates drop and restrictions are eased, due to the already precarious supply of food, and preferential tendencies of large supermarkets to supply metro areas before rural areas [40]. It is possible that the availability of food for some rural regions may have been buffered from the effects of the pandemic in comparison to urban areas, especially where strong, local, and resilient food systems existed. For example, regions where food is grown, sold, and consumed in the same area could potentially have overcome the instability with longer supply chains felt across the world [55]. This may be especially true in rural regions where concern about the spread of the COVID-19 virus was smaller, given that research has shown that higher transmission of the virus significantly increases the likelihood of food access challenges in countries with partial and full lockdown [56].

Challenges related to access to food, including physical and financial access in rural areas, was reported in most of the included studies in our review. During the COVID-19 pandemic across high-income countries, there was lower availability of food, fewer shops open for business in rural areas and media reports of increased price of foods in response to increased consumer demand [57]. These challenges have further infringed upon the ability of rural residents to buy enough food to meet their needs. Interestingly, some studies in our review identified that rural regions were less likely than urban areas to have reduced access to food, especially related to transportation. Indeed, lower reductions in mobility were documented in rural and remote areas compared with urban and metropolitan regions in high-income countries [58], which may have been due to variations in the restrictions in these countries, but may also relate to the need to travel long distances to buy food, and fewer opportunities to purchase food online in rural settings. In rural areas, the higher rates of producing and consuming one’s own food, due to the capacity and tendency to do so, have been suggested to partially buffer issues with food supply and may be somewhat protective against food insecurity during extreme events [56]. Increases in home gardening for food during the pandemic have also been reported [59]. Interestingly, a positive outcome related to food access was reported in our review, with the sudden increase in the delivery of online food shopping opportunities which were not available to rural people prior to the pandemic [36]. The evaluation and maintenance of these successful strategies at improving food access, across differing levels of remoteness and geographical isolation, may help in overcoming pre-pandemic issues with food access for some rural populations going forward, particularly those most at risk of food insecurity who are likely to benefit from these services.

The changes in food availability and access also impacted on the utilization of food and food consumption across the world [60]. For example, it has been reported that there have been increases in the consumption of non-perishable foods as people shopped less frequently [60]. Furthermore, increases in home cooking and eating were documented in high-income countries as people spent more time sheltering in place at home [61]. These changes to the utilization of food are also evident in our review, and the effect of the pandemic on diet quality indicated that there was a reduction in the consumption of fruits and vegetables in some rural regions. Most studies showed there was no difference between rural and urban studies for the outcomes related to food utilization in our review, which may relate to the fact the rural economies and rural livelihoods often do not operate in isolation from urban areas, but are interwoven in many ways. A focus of future research should be to identify strategies to maintain positive aspects of food utilization in rural regions during times of crisis, such as home cooking and growing food [62] The benefits of gardening for food beyond the production have food have been reported during the pandemic [63]. Further studies could also explore changes to food behaviours related to sustainability, such as changes to food waste. 

While only a minority of studies in our review examined the stability of the rural food supply, the included studies suggested that food supply systems in rural areas were impacted. Overall, it was reported that food systems in high-income countries were responsive to the crisis and no major food shortages occurred [64]. However, our review suggests that not all populations had equal access to sufficient food during the pandemic, and rural areas may have been disproportionately affected. Previous analyses of crisis situations have suggested that for some people, including those who experience socio-economic challenges, the struggle exacerbated by a short-term crisis can be more permanent.

Our review demonstrates that only a small number of studies investigated changes to food access and supply in rural regions of high-income countries specifically. Despite many studies that included rural areas and rural participants, the opportunity to assess rurality as a factor was often not considered, with close to 200 manuscripts being excluded on the basis that they examined only urban areas or did not examine the impact of the pandemic on rural subgroups. This indicates our results are aligned with the report by Mueller et al. [51] that suggested rural people have been excluded from the vast majority of research on the impacts of the COVID-19 pandemic. While this leaves a considerable gap in the literature, it also identifies a clear opportunity for existing datasets on food insecurity-related outcomes to be reanalysed according to measures of rurality. Alternatively, researchers should be encouraged to make their datasets publicly available to assist in a more thorough analysis of collected data. Expanding the literature on this topic would serve as a valuable resource when considering policies and strategies that would protect rural populations against the impacts of future pandemics or similar disasters.

### Strengths and Limitations

This review is the first synthesis of the literature using a systematic review methodology that examines how the COVID-19 pandemic has affected food insecurity in rural regions of high-income countries. A strength of this study is the comprehensive search across a broad range of databases to capture multidisciplinary literature related to food security outcomes. The limitations of our review include that there was only a relatively small number of heterogeneous studies meeting the inclusion criteria, which limited our synthesis and precluded a meta-analysis. Furthermore, publication bias may be possible, where studies with neutral or negative results, especially regarding effects in rural subgroups, may not be published or reported, skewing our results. As most of the included studies determined food insecurity using surveys via the Internet, likely due to social distancing restrictions, this means studies could have underrepresented the experience of some rural and remote communities with limited Internet access. It is also possible that the differences in methods of gathering data from participants (such as by telephone or the Internet) limits our ability to confidently compare the results of the included studies. Lastly, only a minority of studies compared the prevalence of food insecurity during the pandemic to food insecurity statistics prior to the pandemic, which limits our ability to determine the true extent to which the pandemic was responsible for the high levels of food insecurity documented in the included studies.

## 5. Conclusions

In conclusion, our review results show that the prevalence of food insecurity in rural populations of high-income countries ranged from 15% to 95%, with food insecurity frequently being significantly higher in rural regions. Rural populations often had lower availability and access to food during the pandemic and experienced multiple changes to food utilization such as reduced diet quality and food safety. Despite the documented increase in food insecurity across high-income countries throughout the COVID-19 pandemic, there is a lack of high-quality data on the prevalence of food insecurity or the domains of food insecurity in rural regions to inform priority areas for intervention at a global scale. Given the disproportionate burden of food insecurity in some rural regions, including additional challenges with food access and supply, the results of this review may be useful for informing region-specific mitigation strategies to reduce the impact of the current pandemic and future global events on food insecurity. Such strategies could include increasing the resilience of rural regions to food supply shocks, through shortening food supply chains and strengthening local food systems [65]. Further research is needed to better characterize the impacts of the COVID-19 pandemic on both urban and rural food insecurity, and there is substantial potential for researchers to analyse existing datasets collected during the pandemic to further examine food security outcomes in rural areas. Such data may be able to adequately inform policies that prepare rural regions for crises in the future. Given the larger proportion of research in urban areas, further research in rural areas is needed to reveal what the optimal recovery pathway could be and how to most effectively promote food security in rural areas specifically, as theyhave been affected differently and will recover differently from the COVID-19 pandemic than urban populations.

## Figures and Tables

**Figure 1 ijerph-19-03235-f001:**
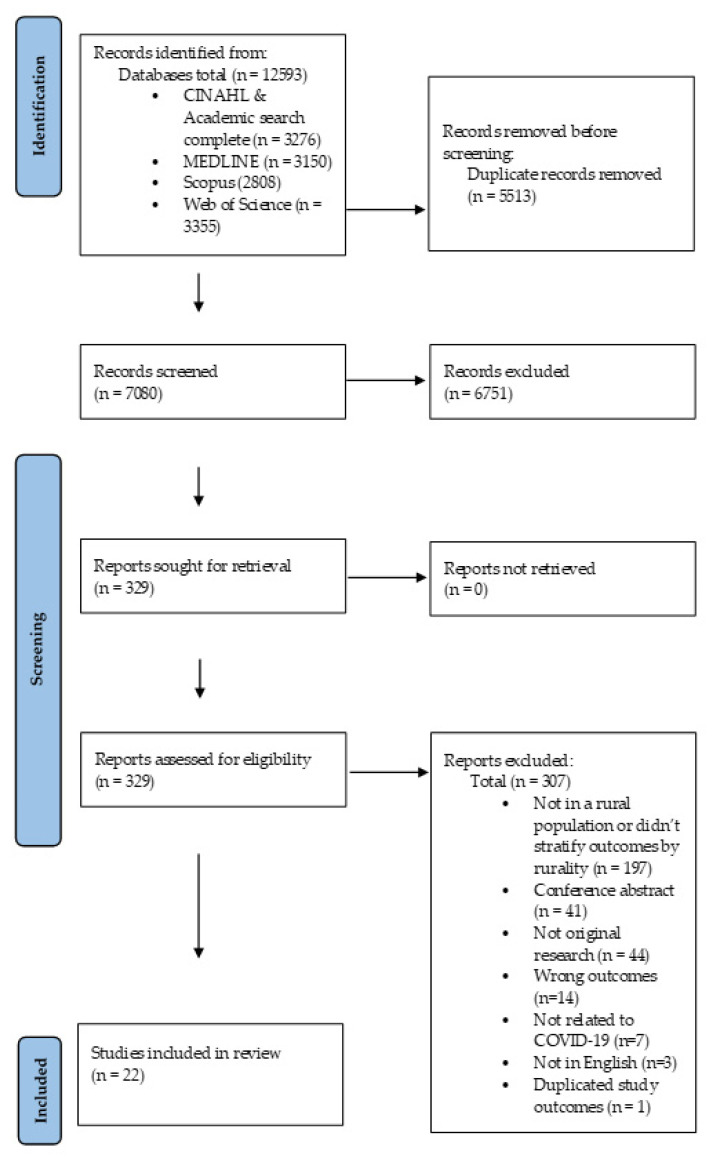
PRISMA flow diagram of search strategy resulting in included studies.

**Table 1 ijerph-19-03235-t001:** Overview of the inclusion and exclusion criteria that guided the study screening.

Section	Criteria	Include If:
Language	Publication reported in English	Yes
Design	Observational studies including prospective and retrospective cohort and cross-sectional studies; or baseline data from intervention studies.	Yes
	Qualitative research including in-depth interviews, focus groups, ethnographic research, content analysis and case studies.	Yes
	Follow-up data from randomised or non-randomised trials, case reports, reviews, editorials, letter to the editor	No
Population	Any age	Yes
	Those living in rural or remote communities as classified by any regional or remote scales	Yes
	Urban dwelling populations, or both urban and rural dwelling populations that have not been stratified by rurality	No
Content	Food security status, including the prevalence of food insecurity, as determined by any valid and reliable screening tool at an individual or population level	Yes
	The experience of food insecurity or hunger	Yes
	Availability of food, including concepts of panic buying, food hoarding and food transport issues	Yes
	Physical access to food such as restrictions on shopping, closure of food outlets and loss of public transport.	Yes
	Financial access to food such as higher food prices, loss of income and lack of social support during the COVID-19 pandemic	Yes
	Utilization of food, including challenges and opportunities around the skills and knowledge surrounding food and food preparation throughout the pandemic	Yes
	Stability of the food supply such as disruptions to the labour or transport needed to maintain the food supply, including apparent consumption data.	Yes
Access	Full-text article accessible	Yes

**Table 2 ijerph-19-03235-t002:** General study overview of included studies, including a checkbox of whether they explored the prevalence of food insecurity, access to food, availability of food, utilization of food or the stability of the food supply.

Reference	Author and Year	Setting	Rural Sample Size	Key Demographics	Food Security Outcome Assessed				
					Food Insecurity Prevalence	Access	Availability	Utilization	Stability
**Quantitative Studies**									
[45]	Giacoman et al., 2021	Adults in rural, regional, and urban Chile	*n* = 504; 11.4% of sample	Total sample demographics not presented	√				
[39]	Kent et al., 2020	Households in Tasmania, Australia. During lockdown.	*n* = 305; 28% of sample	77% female, 68% aged 46+ y; 67% had a university education	√				
[38]	Niles et al., 2020	Households in Vermont, USA	*n* = 1649; 59% of sample classified as rural dwelling	79% female; 96% white; 65% had a university education	√	√	√	√	
[26]	Parekh et al., 2021	Households in rural, suburban and urban USA	*n* = 1766; 31.5% of total sample rural; *n* = 439; 30.2% of sample with children	Households with children (62% female; 60% 40–59 years old; 70% employed) and without children (58% female; 44% 40–59 years old; 56% employed);	√				
[41]	Men et al., 2021	Adults in Canada	*n* = 692; 15.7% of sample rural dwelling	50.8% were female; 48% household with children, 43% immigrants; 50% not working	√				
[32]	Steimle et al., 2021	Socioeconomically disadvantaged parents and their elementary school-aged children in rural Pennsylvania, USA	*n* = 272 rural families	Parents (90% mothers; mean age = 35 years) youngest child aged 4–11, 49% female	√			√	
[30]	Mui et al., 2021	Adults in rural and urban USA	*n* = 385 rural participants; 19% of total sample	Total sample demographics not presented for rural group	√	√	√		
[37]	Niles et al., 2021	Households in Vermont, USA	*n* = 600 participants from ‘A rural state’ but rurality of participants not presented	43.8% were aged 55 y+; 67% female	√	√	√	√	
[28]	Kar et al., 2021	Store data in Franklin County, OH, USA. During and after the state-wide stay-at-home period	7 stores in rural areas	Store characteristics included number of employees, sales, volumes. Characteristics of travellers included average trip length, % white and % income <$50 k/year		√			
[31]	Sherbuk et al., 2020	Patients at a HIV/AIDS clinic in the nonurban southern USA who had low income	*n* = 170 (total sample)	53.5% were men, 58.8% were black, and 61.2% reported that their income fell below the federal poverty line		√			
[46]	Simmet et al., 2021	Food banks in Germany throughout the pandemic	*n* = 18 (4.5%) were rural communities (<500 inhabitants) *n* = 196 were small tows (5000–19,999 inhabitants)	Total sample demographics not presented			√		
[33]	Taylor et al., 2021	Farmers markets in Michigan, USA	*n* = 19 (20%) of farmers markets were in rural areas	Rural markets had a mean of 189 customers/week and 12.4 years in current location. Mean age of market managers was 55.4 years			√		
[44]	Cohn-Schwartz et al., 2021	Adults in rural and urban Israel	*n* = 92; 8% of sample were rural	Adults aged 50+, mean age 63 years, 47% women.				√	
[47]	Jordan et al., 2021	Adults; international; 62 different countries	*n* = 224; 21.4% of respondents were rural	77% were female, 62% aged between 20 and 39 years. Study also explored influence of perceive price on intake but did not report by rurality.				√	
[43]	Sidor and Rzymski	Adults not working regularly in Poland during lockdown (under stay-at-home orders)	*n* = 216; 19.7% of sample were rural	Of total sample: mean age 27.7 (SD = 9.0), 1043 (95.1%) female. 10% unemployed, 47.2% students and 42.8% full time workers. 51.7% tertiary educated				√	
[29]	Luckstead et al., 2021	Adults, low-skilled domestic workers, USA	Survey 1: *n* = 612 respondents Survey 2: *n* = 1036 total respondents; proportion of rural respondents not reported at either timepoint	Not reported; but respondents likely to have income below USD 50,000, without a college degree, and who are below the retirement age of sixty-five					√
**Mixed-Methods Studies**									
[34]	Barr et al., 2021	Adults in Kentucky, USA (a largely rural state)	*n* = 92 emergency food program recipients	72% female, mean age 43.5 ± 15 years, 37% white	√	√			
[27]	Jackson et al., 2022	Adults in rural and urban USA	*n* = 71 (19.7%) of sample	Adults aged 18–78, 52% middle aged, 51.1% female	√			√	
[42]	Snuggs et al., 2021	Adults in UK	*n* = 50; 18.8% of total sample were rural	Of total sample, 208 (86.7%) female; 213 (88.9%) lived in close proximity to a supermarket. *n* = 215 (89.6%) responsible for food shopping in their household				√	
**Qualitative Studies**									
[35]	Barr et al., 2021	Laurel County, Kentucky, USA	*n* = 17	Mean age 54.9 ± 12.6 years		√		√	
[36]	Pyle et al., 2021	A single neighborhood in Oconee County, South Carolina, USA	*n* = 14	55% female, 65% white, 58% high school graduate		√			
[40]	Whelan et al., 2021	Regional community in Victoria, Australia	*n* = 3 supermarket managers, *n* = 33 customers	55% female, did not report other statistics on participants		√	√		√

**Table 3 ijerph-19-03235-t003:** Overview of included studies that reported the prevalence of food insecurity in the respective study settings.

Reference	Method of Data Collection	Food Security Outcome Measures	Analysis Method Used	Interpretation of Results or Key Finding Relating to Rurality Only
**Prevalence Studies**				
[45]	Telephone survey; country-wide	UN FAO Food Insecurity Experience Scale; eight questions; recall period of 30 days	Weighted descriptive statistics; multinomial logistic regression model	53.5% of people in rural Chile were food insecure (combining mild (27.2%) and moderate-severe food insecurity (26.3%)), an increase from 29.9% pre-COVID-19 (mild = 15.7%, moderate-severe 14.2%) ***p* < 0001**. This was higher compared to 45.9% of people in metropolitan regions (mild = 24.6%, moderate-severe = 21.3%) ***p* = 0.001**.
[39]	Online survey; state-wide	U.S. Household Food Security Survey Module: Six-Item Short Form with recall period of 30 days	Descriptive statistics; univariate and multivariate binary logistic regression	33% of rural respondents in Tasmania, Australia were classified into marginal, low and very low food security groups compared to 23% of their urban-dwelling counterparts. After adjusting for other characteristics, authors reported an 82% increase in experiencing food insecurity among respondents in rural areas (AOR: 1.82; SE = 0.34; 95%CI [1.28, 2.62]; *p* = 0.001)
[38]	Online survey; state-wide	U.S. Household Food Security Survey Module: Six-Item Short Form with recall periods “in the year before the coronavirus outbreak” and “since the coronavirus outbreak.”	Kruskal–Wallis tests, Wilcoxon rank sum tests, *t*-tests, and one-way analysis of variance (ANOVA) tests, logistical regression model	***Food insecurity during COVID-19 was 25.1% for rural households, which was higher compared to 21.6% for metropolitan residents (p = 0.035). This consisted of 9.1% of non-metropolitan residents who were newly food insecure during COVID-19 and 16.0% who were food insecure pre-COVID-19.*** In a multivariate model, when comparing newly food insecure to consistently food-insecure households, urban dwelling respondents did not have significantly different risk compared with rural dwelling respondents (B = 0.134: SE = 0.199; 95%CI [0.523–0.255]; *p* = 0.499). When comparing newly food-insecure to consistently food-insecure households, urban-dwelling respondents did not have significantly different risk compared with rural dwelling respondents (B = 0.136: SE = 0.177; 95%CI [0.212–0.484]; *p* = 0.443).
[26]	Online survey; country-wide	U.S. Household Food Security Survey Module: Six-Item Short Form with recall period of 90 days	Descriptive statistics; multivariable logistic regression	***For rural-dwelling households, 15.3% were food insecure, comprising 8.3% who were food insecure and 7.0% with very low food security. This was compared to urban and suburban households with 14.5% FI who had significantly lower rates of very low food security (5.6%, p = 0.041).*** In multivariate analysis for the whole study sample and sample “households with children”, a rural dwelling was not significantly different to suburban areas for food insecurity, adjusting for sex, age, race, region, employment, marital status, education, income, number of people in household (whole study sample AOR: 0.91; 95% CI [0.73–1.12]; sample ”households with children” AOR: 0.94; 95%CI [0.64–1.38])
[34]	Online survey; emergency food program recipients	Hager two-item screener	Descriptive statistics	95% of the sample (participants of a rural emergency food program) were classified as food insecure.
[37]	Online survey; convenience sample of Vermont households from August and September 2020	U.S. Department of Agriculture’s (USDA) Household Food Security Survey Module: Six-Item Short Form, recall periods both “in the year before the coronavirus outbreak” and “since the coronavirus outbreak.	Multivariate logistic regression	29% (*n* = 169) of respondents (and their households) were classified as food insecure since the onset of the COVID-19 pandemic.
[32]	Daily text-messaged surveys of families in a food assistance program during and after school closures located in rural Pennsylvania	Four daily survey questions assessed families’ levels of FI, all adapted for daily use from the Current Population Survey Food Security Supplement	Multilevel, mixed-effects models	For families with children in rural Pennsylvania, all indicators of daily FI significantly increased when schools closed in their region, and gradually decreased in the months that followed. The mean sum of FI question increased from 0.77 before closures to 0.84 after closures (*p* < 0.001). Before and after school closures, the food insecurity (%) was reported for Worry about running out of food (33.5 vs. 38.7%, *p* < 0.01) Parent ate less than should (22.5 vs. 22.2%) Child ate less than should (9.95 vs. 10.0%) Parent or child skipped meal (11.2 vs. 12.8% *p* < 0.001)
[27]	Online, cross-sectional survey	Food insecurity determined using the Hager 2-item food insecurity screener with a recall period of “since the COVID-19 pandemic” began.	Multivariate analysis of covariance (MANCOVA) followed by pairwise univariate tests	40% of participants reported food insecurity. Rurality was not significantly associated with food insecurity.
[41]	Online survey, Canada wide	Household food insecurity in the past 30 days determined by a six-item questionnaire adapted from the 18-item questionnaire that is routinely used to monitor 12-month food insecurity in Canada	Prevalence of outcomes *t*-tests, two-part regression	14.9% of rural dwelling Canadian adults were food insecure, ***significantly higher than for non-rural adults (11.3%, p = 0.017).*** Greater food insecurity in urban versus rural respondents (unadjusted). In an adjusted model, households with children rural residents were more likely to be food insecure than urban households without children, compared with urban residents, respectively.
[30]	Online survey, USA Country-wide	Food insecurity was determined by adapting the 2-item Hunger Vital Sign screening tool with a recall period of the past 30 days of the COVID-19 pandemic.	Chi-square tests	Food insecurity was significantly higher in rural adults (40.5 %; *n* = 156) in the USA, at 62.3 % in urban areas and 36.7 % in suburban areas (*p* < 0.001). A higher proportion of food-insecure adults in rural regions acquired food from supercentres (61.5%; 95% CI [50.4%,72.5%]; *p* < 0.05), than food-insecure adults in suburban areas.

Data presented in bold and italics have been calculated by the review authors from data presented in tables and figures.

**Table 4 ijerph-19-03235-t004:** Overview of included studies that reported the impact of the COVID-19 pandemic on the availability of food in rural areas in the respective study settings.

Food Availability				
Reference	Method of Data Collection	Outcome Measures	Analysis Method Used	Interpretation of Results or Key Finding Relating to Rurality Only
[38]	Online survey; state-wide	Close-ended question developed for the study asked respondents if food was unavailable to them.	Kruskal–Wallis tests, Wilcoxon rank sum tests, *t*-tests, and one-way analysis of ANOVA	Food-insecure respondents were more likely to report that food was unavailable to them during the pandemic than food-secure households, regardless of rurality.
[30]	Online survey; USA wide	One question of food availability in food in retailers	Chi square test	35% of rural participants reported that there was limited availability of food in retailers, compared to urban dwelling (~40%) and suburban (~30%) areas.
[46]	Online survey of food banks in one organization	Days of operation during the COVID-19 pandemic	Chi square test	Of the 401 food banks for which data were available, 58.6% were closed at some point from 16 March to 3 May 2020. On average, food banks were closed 48.1 days (SD 28.7). There were no differences between closed and open food banks concerning the size of the municipality the food bank was located in.
[33]	Telephone or self-administered surveys of Farmer’s Market managers	Participation in subsidized nutrition programs to reduce FI and Impact of COVID-19 on operations	Descriptive statistics	Rural farmers markets: Were more likely to make a profit, with 57.9% of the rural markets making a profit, compared with only 35% in urban clusters and 30.3% in urban areas. Less likely to report a reduction in staff than markets in urban areas. Less likely to report that the number of customers declined. In urban areas it was 43.8% compared to 31.3% of the rural markets.
[40]	Focus groups and group model building	Group model building was undertaken to map issues impacting on the food supply and consumer behaviour	Thematic analysis and causal loop diagrams to describe the system	Rural supermarket managers described ‘empty shelves’ due to panic buying, product unavailability, and community fear generated from the media/mixed messaging. Customers reported fear of not being able to access the food they needed during lockdowns.

**Table 5 ijerph-19-03235-t005:** Overview of included studies that reported on the impact of the COVID-19 pandemic on access to food in rural areas in the respective study settings.

Food Access				
Reference	Method of Data Collection	Outcome Measures	Analysis Method Used	Interpretation of Results or Key Finding Relating to Rurality Only
[37]	Online survey; convenience sample of Vermont households from August and September 2020	Thirteen home food procurement variables developed about home food procurement (local food, gardening, fishing, foraging, hunting, livestock, and canning) examining current practices and changes during the COVID-19 pandemic,	Descriptive statistics	A third of all respondents (34.5%; *n* = 205) engaged in home food procurement activities during the first 6 months of the COVID-19 pandemic. The activities included gardening (34.7%), followed by canning (23.5%) and fishing (10.2%). Most people who engaged in home food procurement activities, 51.8% (*n* = 128), performed at least one home food procurement activity more intensely since the COVID-19 pandemic began.
[31]	Screening patients during standard care. Developed a database of food bank/home delivered meals services provided by the clinic during April 2020 and the preceding 12 months (March 2019 to February 2020).	The variables used to assess food insecurity included: change in employment among patients at risk based on self-report; food support provided through gift cards or delivery of food boxes.	Descriptive statistics	Support for food services increased 66% during April 2020, from 131 average monthly services to 218 services. Home-delivered meals were the most common source of support for patients.
[32]	Daily text messaged surveys of families in a food assistance program during and after school closures located in rural Pennsylvania	Weekly frequency of receiving a “Power Pack” and frequency of use of “Grab and Go” meal options.	Multilevel, mixed-effects models	Families who experienced a greater increase in food insecurity during school closures were more likely to rely upon emergency food relief parcels (“power pack”). Using the power pack services was associated with greater recovery from food insecurity throughout the pandemic. Use of Grab and Go meals was not associated with changes in food insecurity.
[38]	Online survey; state-wide	Close-ended questions developed for the study explored food access challenges and concerns; use of food assistance programs	Kruskal–Wallis tests, Wilcoxon rank sum tests, *t*-tests, and one-way analysis of variance (ANOVA)	Food-insecure households reported more food access challenges compared to food-secure households, including trouble affording food, getting food through a food pantry, and which services were available to help getting food, but this was not affected by rurality.
[34]	Online survey; emergency food program recipients	Open-ended response survey data	Thematic analysis	An emergency food program help alleviate issues of food access (physical and financial) during the pandemic, due to reduced income by “keeping food on the table” and “reducing the frequency of leaving the house”.
[35]	Focus groups; community residents	Exploring experiences in the changes to the food environment during the pandemic	Grounded theory	Respondents described how COVID-19 increased emergency food assistance while other health resources (such as through a library) were restricted. Positive experiences were found with the expansion and utilization of online food ordering, which increased access to food.
[36]	In-depth interviews; community residents	Exploring experiences and the effects of the crises (disasters and pandemics) on community members’ access to food.	A post-positivist theoretical frame	Rural respondents described how food insecurity existed prior to COVID-19, but was exacerbated by the pandemic and disasters in the rural community. Others faced only short-term food insecurity depending on their social networks. A major recurring issue was the scarcity of healthy foods and an increase in food costs during the pandemic.
[40]	Focus groups and group model building	Group model building was undertaken to map issues impacting on the food supply and consumer behaviour	Thematic analysis and causal loop diagrams to describe the system	The nearest larger grocery retailers for one rural community were approximately 70 km by road, inaccessible to many residents during ‘lockdown’. Fear of COVID-19 and of not being able to access food drove panic buying in a rural community.
[28]	Changes in food access based on observed travel data for all grocery shopping trips during and after the state-wide stay-at-home period.	Data in this study were store locations and characteristics store visits (weekly count of customers) assumed characteristics of incoming shoppers as inferred from their origins, and characteristics of store locations	Variables explored was difference in average weekly store visits to during lockdown and initial reopening phases of the pandemic.	During lockdown, traffic declined 2.5 times more in urban stores than those located in rural areas. Stores in urban area experienced a decline in traffic 7.5 times greater than that of a store located in a rural area during the initial reopening phases.
[30]	Online survey	Barriers to food access and transport mode to obtain food	Chi-square tests	Transportation as a barrier did not vary significantly between rural and urban regions. However, food-insecure adults in urban areas faced more barriers to food access and issues obtaining culturally appropriate foods than those in rural areas

**Table 6 ijerph-19-03235-t006:** Overview of included studies that reported the impact of the COVID-19 pandemic on the utilisation of food in rural areas in the respective study settings.

Utilization of Food				
Reference	Method of Data Collection	Outcome Measures	Analysis Method Used	Interpretation of Results or Key Finding Relating to Rurality Only
[38]	Online survey; state-wide	Close-ended question developed for the study asked respondents about concerns of food safety.	Kruskal–Wallis tests, Wilcoxon rank sum tests, *t*-tests, and one-way ANOVA	Food-insecure respondents were more concerned about the safety of food during the COVID-19 pandemic than food-secure households, regardless of rurality.
[37]	Online survey; convenience sample of Vermont households from August and September 2020	Authors developed new questions to measure perceived change in fruit/vegetable and red meat/processed meat consumption since the onset of the COVID-19 pandemic.	Descriptive statistics	Of the sample of rural adults, nearly one in four (23.3%) respondents indicated that ate fewer fruits and vegetables during the pandemic as compared to before, most (65.5%) reported eating the same as before COVID-19, and 11.2% reported eating more. A quarter (25.9%) of respondents reported eating less red and/or processed meat since the start of the COVID-19 pandemic, and 7.9% reported eating more.
[44]	Telephone surveys; country-wide	A single item-question about stocking up food for emergency.	Descriptive data and bivariate analyses	Rural adults were less likely to report stocking up on food than adults living in urban localities.
[35]	Focus groups; community residents	Exploring changes to the food environment during the pandemic	Grounded theory	Rural respondents described how COVID-19 changed the home food environment, including spending more time cooking and eating at home, and increased produce gardening.
[43]	Online cross-sectional survey	Change in food concussion during quarantine; Frequency of consumption of selected food products, Frequency of breakfast consumption during quarantine; Level of fear of contracting SARS-CoV-2 during grocery shopping and contact with food products.	Correlation analysis	Changes in food consumption, snacking, and cooking, eating breakfast, during the quarantine were not differentiated by rurality (*p* > 0.05). Fears regarding contracting COVID-19 from shopping or food sources was not associated with rurality.
[42]	Online survey through social media; close and open-ended questions	The Food Choice Questionnaire and The Family Mealtime Goals Questionnaire. Open-ended questions.	Repeated measures ANOVA	There was no significant effect of suburban/rural location on any of the food choices made by participants in the study.
[27]	Online survey using open-ended and close-ended questions	Diet quality was measured using the dietary screener questionnaire (DSQ), which was modified to assess intake over the past week. Individual nutrient intakes were then combined into an overall measure of diet quality	Multivariate analysis of covariance (MANCOVA) followed by pairwise univariate tests	Over 60% of participants scored a 2 or lower on the 6-point scale of diet quality. Rurality was not linked to dietary quality. However, social connection and changes in dietary behaviour occurred during the pandemic, with food-insecure adults reporting a reduction in diet quality.
[47]	Online survey.	Changes in food quantity, fruit and vegetable consumption as a result of pandemic restrictions. Study also explored the influence of perceive price on intake using open-ended questions but did not report by rurality for this outcome.	Binary logistic regression models and Poisson regression models were calculated to evaluate changes in consumption patterns and to test associations with COVID-19 related factors	The overall effect of living environment was not a significantly influencing vegetable. Regarding consumption, however, among residents of ‘small towns’ (not defined clearly as rural in this study), 20% more (compared to mega cities) reported either an increase/decrease in vegetable intakes. Living in a small town was associated with a reduction in diversity in each of the five vegetable groups reported to be consumed.

**Table 7 ijerph-19-03235-t007:** Overview of included studies that reported the impact of the COVID-19 pandemic on the stability of the food supply rural areas in the respective study settings.

Stability				
Reference	Method of Data Collection	Food Security Outcome Measures	Analysis Method Used	Interpretation of Results or Key Finding Relating to Rurality Only
[29]	Online survey of a representative sample of low-skilled domestic workers’ attitudes; USA country wide	Perceived importance of agriculture food production and concern about having a food shortage due to the effect of COVID-19 amid the coronavirus crisis.	Descriptive statistics; Logistic and Order Logistic regression	On average, the total sample of respondents perceived that agricultural production was more important than before the crisis and were more concerned about a food shortage than prior to the pandemic. There was no difference between rural and urban respondents regarding the perceived importance of agricultural production and concern for food shortage.
[40]	Focus groups and group model building	Group model building was undertaken to map issues impacting on the food supply and consumer behaviour	Thematic analysis and causal loop diagrams to describe the system	Supermarket managers described an unpredictability in consumer behaviour as well as supply chain issues as a result of COVID-19, that led to a lack of stability in the local food supply. The supply chain struggled to re-orient itself in a tight time frame.

## Data Availability

Data supporting this review are available from the authors upon reasonable written request including the template data collection forms; data extracted from included studies; or any other materials used in the review.

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
