# Peer review of "The Impact of the COVID-19 Pandemic on Rural Food Security in High Income Countries: A Systematic Literature Review"

_ijerph, 2022, doi:10.3390/ijerph19063235_

Round 1

Reviewer 2 Report

Specific points

In general, this study contains research on: differences in food availability, problems of access to food and making а stockpiling, food shopping opportunities, food supply stability, food shortage as well as sociodemographic differences issues with food supply of rural, in Covid-period.

In my opinion, this study does not contain data on healthy food and quality of nutrition . And that should be paid attention to and corrected everywhere in the text where the terms "healthy food",   "healthy diet" and "quality of nutrition" are mentioned.

In my opinion, parts of the text under subheadings: 3.1 Study selection and 3.2. Description of studies, belong to the section Material and methods. Namely, the authors do not present the results here on "The impact of the COVID-19 pandemic on rural food security ...". In fact, they explain the materials they studied here.

L:229-251: Same comment as above

L:491-492: I think there should be more precision in the use of certain terms. For example, in the sentence: "These changes to the utilization of food are also evident in our review, but the effect of the pandemic on diet quality have been mixed, ..." - one cannot speak of "quality of nutrition". The quality of the diet would include an assessment of the intake of certain nutrients when consuming food. Namely, the authors present a study on the availability of food, not on the quality of food. I think the authors need to be much more precise in that regard. Please, correct this in the text. For example, this is repeated in L:498, 499.

L:492-494: Same comment as above. Need to be precise in use of terms. For example, in the sentence: “…with some studies showing an increase in healthy food due to home cooking and others showing a reduction in consumption of healthy food (such as vegetables) in rural regions.” How do authors know that food prepared at home is healthy? Do they think that this applies to any food prepared at home, in general? In addition, it does not mean that every vegetable is healthy; for example, if it contains pesticides and mycotoxins (due to poor cultivation and storage) - it will not be healthy. Namely, in order for a food to be labeled “healthy”, the composition of its macro- and micronutrients must be given. And that was not the goal of this study.

L:504-505: Same comment as above. “However, our review suggests that not all populations had equal access to sufficient, healthy food during the pandemic,…”- on what basis is it claimed that it is about greater availability of “healthy food”?

  1. Conclusions: In fact, no conclusions have been drawn from this study. it was pointed out that valid data are missing and what significance this study could have. I think that the research presented here should be "rounded off" in the conclusion. So, what was the availability and access to food, what was the possibility of buying food, food shortages, stability of food supply ... in rural regions? I think that the conclusion of this research should be rewritten.

________________

All my suggestions are for improving the manuscript. I hope all the suggestions are clear.

Best regards

Reviewer 3 Report

The review paper is written well and highlights important issues that affected all regions of the world during pandemic regarding food security. This paper is mainly focusing on the rural areas and using published paper (review) to draw some conclusions.

There are some areas in the paper that need more clarification and explanations.

These are general comments, beside the fact the sample size of 22 articles is small and may not draw significant findings. 

a) Did authors look at the prepandeimc and find out if there were any significant changes regarding the food security during pandemic? As rural areas, especially remote areas,  are generally have less access to food. It was only comparison between rural and urban.

b) Is there any reason that only high income countries chosen? How about middle and low income? I feel there will be interesting to discover how the other countries were doing.

c) The methods of gathering data from participants were different in each study. There were online, telephone surveys, text messaging, and focus groups. Would these methods are all considered in the analysis the same way? 

d) What were the main causes of food insecurity and if they changed during pandemic? shortage, transportation, higher food prices or other reasons?

c) Figures and table were very good and illustrate the selecting and screening of the studies. Have any statistical analysis were done to determine if the findings were significant although with small sample size may not be conclusive.

d) Final point: this manuscript is for the IJERPH. I would assume that the authors could address some environmental factors in their findings such as consumer behavioral changes such reducing food loss and waste and consuming all the food available to them as there was fear of shortage. 

The local gardening and food production could be promoted to conserve environment and provide food at the time of crisis. 
